# Kerr and Faraday rotations in topological flat and dispersive band structures

Alireza Habibi,[1] Ahmad Z. Musthofa,[2] Elaheh Adibi,[3] Johan Ekström,[1] Thomas L. Schmidt,[1] and Eddwi H. Hasdeo[1,4,*]

[1]*Department of Physics and Materials Science, University of Luxembourg, L-1511 Luxembourg*
[2]*Department of Physics, Brawijaya University, Malang 65145, Indonesia*
[3]*Institute for Advanced Simulation, Forschungszentrum Jülich, 52425 Jülich, Germany*
[4]*Research Center for Physics, National Research and Innovation Agency, South Tangerang 15314, Indonesia*

Integer quantum Hall (IQH) states and quantum anomalous Hall (QAH) states show the same static (dc) response but distinct dynamical (ac) response. In particular, the ac anomalous Hall conductivity profile $\sigma_{yx}(\omega)$ is sensitive to the band shape of QAH states. For example, dispersive QAH bands shows resonance profile without a sign change at the band gap while the IQH states shows the sign change resonance at the cyclotron energy. We argue by flattening the dispersive QAH bands, $\sigma_{yx}(\omega)$ should recover to that of flat Landau bands in IQH, thus it is necessary to know the origin of the sign change. Taking a topological lattice model with tunable bandwidth, we found that the origin of the sign change is not the band gap but the Van Hove singularity energy of the QAH bands. In the limit of small bandwidth, the flat QAH bands recovers $\sigma_{yx}(\omega)$ of the IQH Landau bands. Because of the Hall response, these topological bands exhibit giant polarization rotation and ellipticity in the reflected waves (Kerr effect) and rotation in the order of fine structure constant in the transmitted waves (Faraday effect) with profile resembles $\sigma_{yx}(\omega)$. Our results serve as a simple guide to optical characterization for topological flat bands.

## I. INTRODUCTION

The internal structure of electron wave functions in solids can lead to unconventional transport phenomena. A quantized Hall conductivity $\sigma_{yx}$ without externally applied magnetic field in quantum anomalous Hall (QAH) materials is one example of such an effect [1, 2]. In close analogy to the integer quantum Hall (IQH) effect, the Berry curvature effectively acts as a magnetic field in momentum space, which globally gives rise to a nonzero Chern number resulting in the quantized $\sigma_{yx}$. In the case of a static (dc) electric field, there exist dissipationless chiral edge modes both in QAH and IQH states as a consequence of the bulk-boundary correspondence.

Exciting IQH states with dynamical (ac) electromagnetic fields causes a rotation of the polarizations of the transmitted and reflected light, respectively known as the Faraday and Kerr rotations [3–8]. The energy-dependent conductivity tensor can be experimentally extracted from these rotation angles. It is known that $\sigma_{yx}(\omega)$ for an IQH state shows resonant behavior marked by a change of sign at frequencies $\omega$ near the cyclotron frequency [9, 10]. However, in QAH systems $\sigma_{yx}(\omega)$ can display different resonant profiles depending on the details of the system. For example, a simple two-band QAH model such as for a gapped Dirac Hamiltonian, gives rise to a resonant profile without sign change at the band gap [6, 11, 12] whereas the four-band models for bilayer graphene show a resonance with sign change at higher energy than the broken time reversal symmetry gap [13]. Focusing on two band models only, one may thus ask if the behavior of $\sigma_{yx}(\omega)$ in QAH systems can reproduce that of IQH states by flattening down the dispersive topological bands.

In this work, we employ a topological lattice model whose bandwidth can be tuned smoothly to capture nearly flat and dispersive bands on an equal footing [14, 15]. Other than the bandgap, $\sigma_{yx}(\omega)$ is determined by a van Hove singularity (VHS) which naturally emerges in this 2D lattice model. This addi-

tional parameter determines the sign change of $\sigma_{yx}(\omega)$ similarly to Ref. [13]. In topological flat bands, the VHS energy coincides with the band gap resulting in a change of sign in the resonance profile. In dispersive bands, in contrast, the VHS energy is higher than the band gap. As a result, the Hall conductivity shows separate resonance and sign change features, respectively, at the band gap and VHS energy. The different features of the Hall conductivity in the cases of flat and dispersive bands, respectively, are reflected in the corresponding Faraday angle $\theta_F$. At low frequencies, $\theta_F$ approaches a universal value $\tan^{-1}(\alpha)$, where $\alpha = e^2/\hbar c \approx 1/137$ is the fine structure constant [6]. At larger frequencies, $\theta_F$ follows the trend of the Hall conductivity with a negative prefactor. In the reflected wave, a giant Kerr angle $(\pi/2)$ below the gap is expected for Chern insulators [6]. Above the gap, the Kerr angle changes sign and decays to zero when approaching the bandwidth.

From a fundamental point of view, our results serve as a one-to-one mapping of IQH states to QAH states. All phenomena that happen in IQH states with magnetic field can also occur in QAH states without magnetic field (but with nonzero Berry curvature). To name a few, the Kerr and Faraday rotations, cyclotron motion [12], and fractional excitation without magnetic field [14–17] can be observed in QAH states. From a practical point of view, these results provide a simple characterization tool to determine the bandwidth or band flatness of topological materials. The experimental observation of flat bands is currently limited only to angle-resolved photo-emission spectroscopy [18, 19]. However, due to the band flatness and the close proximity to the Fermi energy, a flat band is usually difficult to characterize. A convenient platform to study flat bands with nontrivial topology has been realized in magic-angle twisted bilayer graphene in which strong electronic correlations cause a breaking of time-reversal symmetry [20–22].

## II. MODEL HAMILTONIAN AND BAND STRUCTURE

An effective two-band Hamiltonian which can model a system with topological flat bands on a square lattice can be written

* hesky.hasdeo@uni.lu

as [14, 15]

$$\hat{H} = \boldsymbol{\tau} \cdot \mathbf{d}(\mathbf{p}),$$
(1)

where $\boldsymbol{\tau} = (\tau_x, \tau_y, \tau_z)$ is a vector of Pauli matrices and $\mathbf{p} = \hbar\mathbf{k}$ denotes the quasi-momentum. The vector $\mathbf{d}$, which describes the band structure of the material, is given by

$$\mathbf{d}(\mathbf{p}) = \frac{\hbar v}{a} \begin{pmatrix} \cos \frac{p_x a}{2\hbar} \cos \frac{p_y a}{2\hbar} \\ \sin \frac{p_x a}{2\hbar} \sin \frac{p_y a}{2\hbar} \\ b \left( \cos \frac{p_x a}{\hbar} - \cos \frac{p_y a}{\hbar} \right) \end{pmatrix},$$

where $a$ and $v$ denote the lattice spacing and Fermi velocity of electrons, respectively. The $d_y$ component is an even function of $\mathbf{p}$ and indicates time-reversal symmetry breaking because the complex conjugate of $\tau_y$ picks up a minus sign under time reversal. As a result, the bands are topologically nontrivial for $b \neq 0$ and the Chern number is given by $\pm\text{sign}(b)$ for conduction and valence bands, respectively [14]. For $|b| < b_0 = (2\sqrt{2})^{-1}$, the bandgap can be defined as $2\Delta$ with $\Delta = \hbar v(2b)/a$. Diagonalizing $\hat{H}$ gives the energy spectrum

$$\mathcal{E}_\pm(\mathbf{p}) = \pm|\mathbf{d}(\mathbf{p})| = \pm \frac{1}{\sqrt{2}} \frac{\hbar v}{a} \sqrt{1 + 2b^2 \left( \cos \frac{p_x a}{\hbar} + \cos \frac{p_y a}{\hbar} \right)^2 + (1 - 8b^2) \cos \frac{p_x a}{\hbar} \cos \frac{p_y a}{\hbar}}.$$
(2)

The parameter $b$ not only determines the band gap but also the band width. For the particular choice $b = b_0 = (2\sqrt{2})^{-1}$, the flat bands emerge with ratio of the bandwidth to bandgap equal to $(1 - \sqrt{2})/2 \approx 0.2$. These (almost) flat bands are shown in Fig. 1(a) and their energy spectra are given by

$$\mathcal{E}_\pm(\mathbf{p}) = \pm \frac{\sqrt{2}}{4} \frac{\hbar v}{a} \sqrt{\left( \cos \frac{p_x a}{\hbar} + \cos \frac{p_y a}{\hbar} \right)^2 + 4}.$$
(3)

The density of state (DOS) is defined as

$$\rho(\mathcal{E}) = \sum_{\mathbf{p}} \sum_{\eta=\pm} \delta(\mathcal{E} - \mathcal{E}_\eta(\mathbf{p})),$$
(4)

For the flat bands at $b = b_0$, an analytical form of DOS can be obtained as:

$$\rho(\mathcal{E}) = \frac{a}{\hbar v} \frac{4\sqrt{2}}{\pi^2} \frac{|\bar{\varepsilon}|}{\sqrt{\bar{\varepsilon}^2 - 1}} \mathcal{K}(2 - \bar{\varepsilon}^2) \quad (1 < |\bar{\varepsilon}| < \sqrt{2}),$$
(5)

where $\bar{\varepsilon} = \mathcal{E}/\Delta$ denotes the dimensionless energy and $\mathcal{K}$ is an elliptic integral defined in Appendix A Eq. (A11) [23]. The density of states of the flat bands is shown in Fig. 1(b). It becomes singular both at the maximum of the valence band, and the minimum of the conduction band because the saddle-points coincide with the band edges.

Let us contrast this band structure with that of dispersive bands, shown for example for $b = 0.075$ in Fig. 1(c) and (d). The band morphology is similar to that of gapped graphene, but this model is defined on a square lattice rather than on a hexagonal one. At the band edges, the density of states is finite, as expected for parabolic bands in 2D systems. Away from the band gap, the DOS exhibits a VHS as a result of the saddle-point band dispersion, similar to graphene. We will compare the optical conductivities of these different bands in the next section.

## III. OPTICAL CONDUCTIVITY

An electron gas with two sites per unit cell can be described using a pseudo-spin. Matrix elements of this pseudo-spin encode

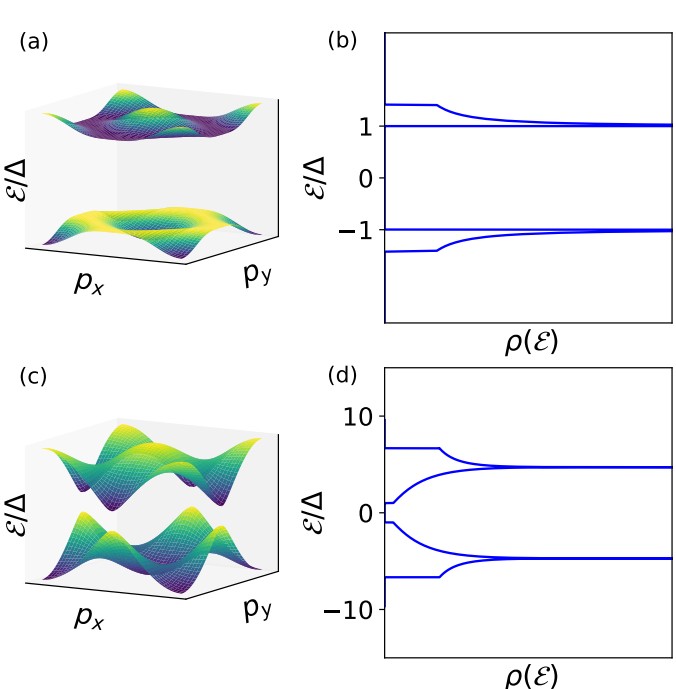

FIG. 1. (a) The flat bands obtained from Eq. (2) for $b = b_0 = (2\sqrt{2})^{-1}$. (b) Density of states (DOS) of the flat bands. (c) The dispersive bands obtained from Eq. (2) for $b = 0.075$. (d) DOS of the dispersive bands. Here, $\Delta = \hbar v(2b)/a$.

the wave function as well as geometric properties that underlie electronic transport [12]. The pseudo-spin is defined as

$$\mathbf{m}_{\mathbf{p}} = \langle \psi(\mathbf{p})|\boldsymbol{\tau}|\psi(\mathbf{p})\rangle,$$
(6)

where $\psi(\mathbf{p}) = (\psi_A(\mathbf{p}), \psi_B(\mathbf{p}))$ is the wavefunction on the $A$ and $B$ sites. The pseudo-spin obeys the Bloch equation,

$$\frac{d}{dt}\mathbf{m}_{\mathbf{p}}(t) = \langle \psi(\mathbf{p})|\frac{1}{i\hbar}[\boldsymbol{\tau}, \hat{H}]|\psi(\mathbf{p})\rangle = \frac{2}{\hbar} \mathbf{d}(\mathbf{p}) \times \mathbf{m}_{\mathbf{p}}.$$
(7)

In the absence of an applied electric field, $\mathbf{m}_{\mathbf{p}} = \mathbf{m}_{\mathbf{p}}^{(0)} = \pm\mathbf{d}(\mathbf{p})/|\mathbf{d}(\mathbf{p})|$ where the $\pm$ signs correspond to the pseudo-spin

of conduction and valence band, respectively. Therefore, the pseudo-spins of the conduction (valence) band are oriented parallel (antiparallel) to $\mathbf{d}(\mathbf{p})$, which in turn leads to $d\mathbf{m}_\mathbf{p}(t)/dt = 0$.

Upon applying an external electric field $\mathbf{E}$, the vector $\mathbf{d}$ shifts up to linear order as $\delta d_j = \mathbf{A} \cdot \partial_\mathbf{A} d_j(\mathbf{p} - e\mathbf{A}/c)$ where $e$ and $c$ are electron charge and speed of light, respectively. Furthermore, the vector potential $\mathbf{A}$ is related to $\mathbf{E}$ via the relation $\mathbf{E} = -\partial_t \mathbf{A}/c$. On the other hand, in presence of an electric field, the pseudo-spin changes to $\mathbf{m}_\mathbf{p}(t) = \mathbf{m}_\mathbf{p}^{(0)} + \delta\mathbf{m}_\mathbf{p}(t)$, so it is no longer aligned with $\mathbf{d}' = \mathbf{d} + \delta\mathbf{d}$. Hence, Eq. (7) predicts a precession of the pseudo-spin. Focusing on linear response to the incident light, Eq. (7) becomes

$$\frac{d}{dt}\delta\mathbf{m}_\mathbf{p}(t) = \frac{2}{\hbar}\Big[\mathbf{d}(\mathbf{p}) \times \delta\mathbf{m}_\mathbf{p}(t) + \delta\mathbf{d}(\mathbf{p}) \times \mathbf{m}_\mathbf{p}^{(0)}\Big], \quad (8)$$

where we dropped the nonlinear term $\delta\mathbf{m}_\mathbf{p} \times \delta\mathbf{d}$.

Considering without loss of generality an incident electric field linearly polarized along the $x$ axis, we have $\mathbf{E}(t) = E_x\, e^{-i\omega t}\hat{\mathbf{x}}$, and obtain

$$\delta\mathbf{d}(\mathbf{p}) = \frac{iev E_x}{2\omega} \begin{pmatrix} \sin\frac{p_x a}{2\hbar}\cos\frac{p_y a}{2\hbar} \\ -\cos\frac{p_x a}{2\hbar}\sin\frac{p_y a}{2\hbar} \\ 2b\sin\frac{p_x a}{\hbar} \end{pmatrix}. \quad (9)$$

In equilibrium the electrons occupy the valence band, so $\mathbf{m}_\mathbf{p}^{(0)} = -|\mathbf{d}|/\mathbf{d}$ and we can compute $\delta\mathbf{m}$ by solving Eq. (8). We obtain

$$\delta m_x = \frac{\delta d_x \left(d_z^2 + d_y^2\right) + \delta d_y \left[-i\left(\hbar\omega/2\right) d_z - d_x d_y\right] + \delta d_z\left[i\left(\hbar\omega/2\right) d_y - d_x d_z\right]}{|\mathbf{d}| \left[\left(\hbar\omega/2\right)^2 - |\mathbf{d}|^2\right]}, \quad (10)$$

$$\delta m_y = \frac{\delta d_x\left[i\left(\hbar\omega/2\right) d_z - d_x d_y\right] + \delta d_y \left(d_z^2 + d_x^2\right) + \delta d_z \left[-i\left(\hbar\omega/2\right) d_x - d_y d_z\right]}{|\mathbf{d}| \left[\left(\hbar\omega/2\right)^2 - |\mathbf{d}|^2\right]}, \quad (11)$$

$$\delta m_z = \frac{\delta d_x \left[-i\left(\hbar\omega/2\right) d_y - d_x d_z\right] + \delta d_y\left[i\left(\hbar\omega/2\right) d_x - d_y d_z\right] + \delta d_z \left(d_x^2 + d_y^2\right)}{|\mathbf{d}| \left[\left(\hbar\omega/2\right)^2 - |\mathbf{d}|^2\right]}. \quad (12)$$

Further details on pseudo-spin dynamics are presented in Ref. [12] and its supplementary information.

The electric field induces both longitudinal and Hall current responses in the system. In fact, up to linear response the optical conductivity $\sigma_{ij}$ connects the electrical current in direction $i$ to an external transverse electric field in the $j$ direction, $\mathbf{J} = \sigma\mathbf{E}$. In the following, we determine both responses by using the pseudo-spins.

### A. Longitudinal conductivity

The longitudinal current is given by

$$J_x(\omega) = \frac{e}{(2\pi\hbar)^2}\int d^2\mathbf{p}\left\langle\frac{\partial\hat{H}}{\partial p_x}\right\rangle. \quad (13)$$

Utilizing the relation $J_x(\omega) = \sigma_{xx}(\omega)E_x$ and calculating the above expectation value for the valence band states, we obtain the interband longitudinal conductivity $\sigma_{xx}$ as

$$\sigma_{xx}(\omega) = -\mathcal{A}\int d^2\mathbf{p}\left(\delta\widetilde{d}_x\,\delta m_x + \delta\widetilde{d}_y\,\delta m_y + \delta\widetilde{d}_z\,\delta m_z\right), \quad (14)$$

where $\mathcal{A} = \frac{ev}{(2\pi\hbar)^2 E_x}$ and $\delta\widetilde{\mathbf{d}} = \frac{\omega}{iev E_x}\delta\mathbf{d}$. In the end, $\sigma_{xx}$ is independent of $E_x$ because $\delta\mathbf{m}_\mathbf{p} \sim E_x$ (see Appendix B for

details). An analytical form can be obtained for the flat bands by setting $b = b_0$. We find the real part of $\sigma_{xx}$ to be given by

$$\text{Re}\,\sigma_{xx}^{(\text{flat})}(\overline{\omega}) = \frac{e^2}{h}\frac{\mathcal{K}\left(2 - \overline{\omega}^2\right) - \left(\overline{\omega}^2 - 1\right)\mathcal{L}\left(2 - \overline{\omega}^2\right)}{\overline{\omega}^2\sqrt{\overline{\omega}^2 - 1}}, \quad (15)$$

for $1 < \overline{\omega} < \sqrt{2}$ and where we used dimensionless frequency $\overline{\omega} = \hbar\omega/(2\Delta)$. Here, $\mathcal{K}$ and $\mathcal{L}$ are elliptic integrals of the first and second kind which are defined in Eq. (A11) and (B7).

We can evaluate the imaginary part of the longitudinal conductivity numerically by using the Kramers-Kronig relation:

$$\text{Im}\,\sigma_{xx}(\omega) = -\frac{2\omega}{\pi}\int_0^\infty\frac{\text{Re}\,\sigma_{xx}(\omega')}{\omega'^2 - \omega^2}\,d\omega'. \quad (16)$$

In Fig. 2, we plot the real and imaginary parts of the longitudinal conductivity in the units of the conductance quantum $e^2/h$.

### B. Hall conductivity

In order to investigate the anomalous Hall current, we use

$$J_y(\omega) = \frac{e}{(2\pi\hbar)^2}\int d^2\mathbf{p}\left\langle\frac{\partial\hat{H}}{\partial p_y}\right\rangle. \quad (17)$$

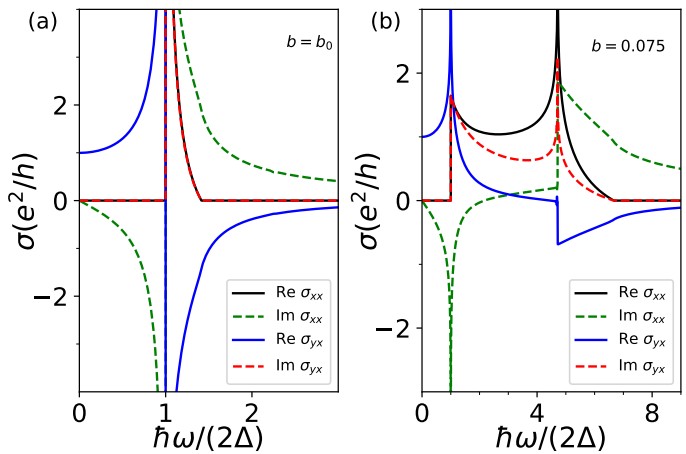

FIG. 2. (color online). Optical conductivity in longitudinal $\sigma_{xx}$ and transverse (Hall) directions $\sigma_{yx}$ for (a) the flat bands $b = b_0$ and (b) the dispersive bands $b = 0.075$.

Computing the expectation value on the valence band states and considering $j_y(\omega) = \sigma_{yx}(\omega)E_x$, the anomalous Hall conductivity can be written as

$$\sigma_{yx}(\omega) = \mathcal{A} \sum_{\mathbf{p}} \left( \delta\widetilde{d}_y \, \delta m_x + \delta\widetilde{d}_x \, \delta m_y + b\sin\left(\frac{p_y a}{\hbar}\right) \delta m_z \right). \tag{18}$$

For the flat bands, we calculate the imaginary part of the Hall conductivity analytically in Appendix C, which gives

$$\text{Im } \sigma_{yx}^{(\text{flat})}(\overline{\omega}) = \frac{e^2}{h} \frac{2 - \overline{\omega}^2}{\overline{\omega}} \frac{\mathcal{K}\left(2 - \overline{\omega}^2\right)}{\sqrt{\overline{\omega}^2 - 1}}, \quad (1 < \overline{\omega} < \sqrt{2}). \tag{19}$$

Using the Kramers-Kronig relation we obtain the real part of $\sigma_{yx}(\omega)$ as

$$\text{Re } \sigma_{yx}^{(\text{flat})}(\overline{\omega}) \tag{20}$$
$$= \frac{e^2}{h} \frac{1}{\pi} \int_{-2}^{2} d\lambda \, \frac{(\lambda - 2)(\lambda + 2)}{(4\overline{\omega}^2 - 4 - \lambda^2)\sqrt{\lambda^2 + 4}} \mathcal{K}\left(1 - \frac{\lambda^2}{4}\right).$$

Details of the derivation are presented in Appendix C. We perform the integration over $\lambda$ numerically. The plots of the real and imaginary parts of $\sigma_{yx}$ in units of $e^2/h$, as a function of frequency $\overline{\omega} = \hbar\omega/(2\Delta)$ are shown in Fig. 2.

### C. Discussion

For $\sigma_{xx}$, the real part is plotted in black lines while the imaginary part is in green dashed lines, for the flat bands in Fig. 2(a), and for the dispersive bands with $b = 0.075$ in Fig. 2(b). The profiles of the real part are more or less similar to the joint DOS which is expected for dissipative interband transition. On the other hand, the imaginary part takes on negative values and flips sign at the VHS. Positive and negative values of Im $\sigma_{xx}$ are related to transverse magnetic (TM) and transverse electric (TE)

surface waves, respectively, in the absence of Hall response [24]. In the dispersive band, see Fig. 2(b), TE surface waves can survive up to an excitation energy $\hbar\omega$ larger than about three times of the gap size.

Next, we describe the properties of the anomalous Hall conductivity $\sigma_{yx}$. Its real part (blue lines of Fig. 2) is quantized to $e^2/h$ at zero frequency, as expected for Chern insulators with the Chern number one. For the flat bands, Re $\sigma_{yx}$ increases and diverges at the band gap. Above the band gap, it changes sign and goes to zero for large frequencies. This profile resembles the Hall conductivity of magnetotransport $\sigma_{yx}^B = \sigma_0^B(1-x^2)^{-1}$, where $x = \omega/\omega_c$. In the quantum Hall regime, $\sigma_0 = Ce^2/h$ [10] with $C$ being the Chern number and in the semiclassical regime, $\sigma_0^B = ne^2/m\omega_c$, $\omega_c$ is the cyclotron frequency, and $n$ and $m$ are the electron density and electron mass, respectively [25].

Although our model does not have a perfectly flat band, its properties nevertheless resemble those of magnetotransport for a flat Landau level. A hint to elucidating this similarity can be traced back to Re $\sigma_{yx}$ of the dispersive band in Fig. 2(b). For the dispersive bands, Re $\sigma_{yx}$ does not change sign at the band gap. It does, however, change sign after it passes the VHS energy. Thus, the flip of sign in Re $\sigma_{yx}$ is related to the position of the VHS. In the flat bands, the position of the VHS matches the band edge, so Re $\sigma_{yx}$ flips the sign for excitation energy $\hbar\omega = 2\Delta$. Nevertheless, the resonant point of $\sigma_{yx}$ remains at the band gap energy and it serves as the cyclotron energy of the anomalous Hall materials.

The imaginary part of $\sigma_{yx}$ coincides with the real part of $\sigma_{xx}$ in the flat bands. Numerically, there is actually a small discrepancy between the two values that makes them hard to distinguish. However, this coincidence is not generally valid for all $\omega$ negative because of the odd symmetry Im $\sigma_{yx}(-\omega) = -\text{Im } \sigma_{yx}(\omega)$ while Re $\sigma_{xx}$ $(\omega)$ is an even function. Indeed Im $\sigma_{yx}$ is distinct from Re $\sigma_{xx}$ in the dispersive bands as shown in black solid line vs red dashed line of Fig. 2(b).

### IV. KERR AND FARADAY ANGLES

Topological bands allow transverse current responses in the presence of an electric field. Thus, such a material can rotate the polarization plane of the reflected and transmitted electric field. The former is known as the Kerr rotation while the latter is called the Faraday rotation. For the simplest geometry, i.e., normal incidence of an electric field $E_0\hat{\mathbf{x}}$ onto a free-standing two dimensional material, the Kerr ($\theta_K$) and Faraday ($\theta_F$) angles can be defined as [26]

$$2\theta_{K,F} = \tan^{-1}\left(\frac{\text{Im}E_s^R(\omega)}{\text{Re}E_s^R(\omega)}\right) - \tan^{-1}\left(\frac{\text{Im}E_s^L(\omega)}{\text{Re}E_s^L(\omega)}\right), \tag{21}$$

and the respective ellipticity angles are defined as

$$\xi_{K,F} = \frac{|E_s^R|}{|E_s^L|}, \tag{22}$$

where the subscripts $s = r, t$ denote the reflected and transmitted components of the electric field for Kerr ($K$) or Faraday ($F$) rotation, respectively. Moreover, the superscripts $R$ and $L$ denote right-circular and left-circular polarized components. We solve

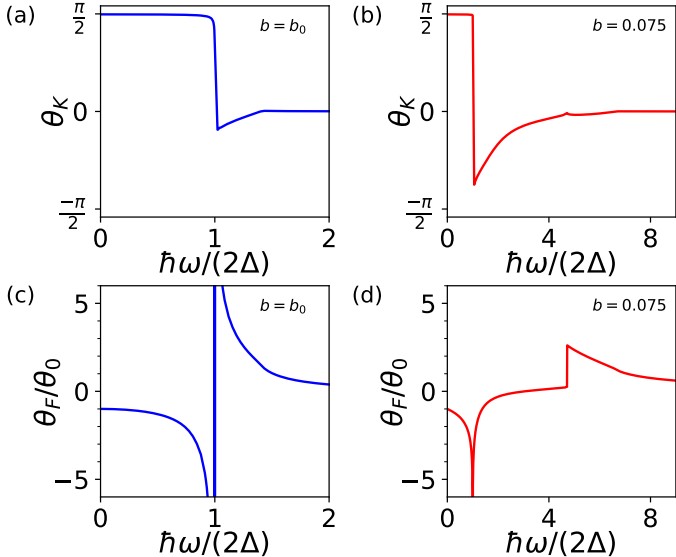

FIG. 3. (color online). Kerr angles of (a) the flat bands and (b) the dispersive bands as a function of excitation energy. Faraday angles of (c) the flat bands and (d) the dispersive bands as a function of excitation energy. We have used $\theta_0 = \tan^{-1}(\alpha) = 7.3 \times 10^{-3}$ rad.

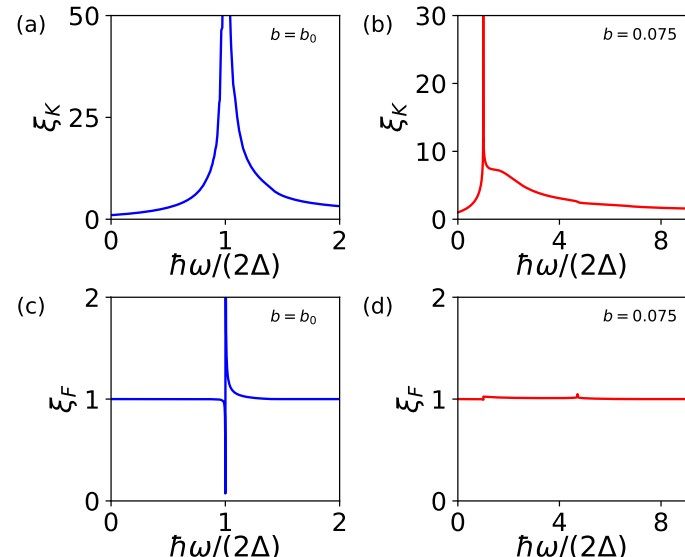

FIG. 4. (color online). *Ellipticity*. Kerr ellipticity of reflected waves for (a) the flat bands and (b) the dispersive bands as a function of excitation energy. Faraday ellipticity of transmitted waves for (c) the flat bands and (d) the dispersive bands as a function of excitation energy.

the Maxwell equation with boundary condition that electric field parallel to the surface is continuous, while the magnetization is discontinous due to surface current. The electric fields are given by [27]

$$E_r^{R,L} = E_0 \frac{\sigma^\mp(\omega)}{\kappa_1 - \sigma^\mp(\omega)}, \tag{23}$$

$$E_t^{R,L} = 2\kappa_1 E_0 \frac{\kappa_1 - \sigma^\pm(\omega)}{\left(\kappa_1 - \sigma^+(\omega)\right)^2 + \left(\kappa_1 - \sigma^-(\omega)\right)^2}, \tag{24}$$

$$\sigma^\pm = \left(\sigma_{xx} \pm i\sigma_{yx}\right)/\sigma_0, \tag{25}$$

where $\kappa_1 = \alpha^{-1} = 137$, $\alpha = e^2/(\hbar c)$ is the fine structure constant, and $\sigma_0 = e^2/h$. The Kerr angle of the flat bands is shown in Fig. 3(a). Below the gap, $\theta_K = \tan^{-1}\left(\alpha^{-1}\right) \approx \pi/2$ as expected for topological insulators [6]. This is because below the gap Re $\sigma_{xx} = 0$ and $\kappa_1 \gg \sigma_{yx}/\sigma_0$. The value of $\theta_K$ below the gap is insensitive to the detailed shape of the band structure as shown in Fig. 3(b) but depends on the incident angle and the dielectric environment, which are not considered here. Above the gap, it changes sign and gradually approaches zero. Excitations above the bandwidth do not give a Kerr rotation. Apparently the large values of $|\theta_K|$ are proportional to the bandwidth as shown in Fig. 3(b) as compared to (a). The profile of the Faraday angle resembles $-\text{Re}\,\sigma_{yx}$ up to a constant prefactor. At zero frequency, $\theta_F = \tan^{-1}(\alpha)$, as has been shown in Ref. [6].

Apart from the rotation of the polarization angles, an anomalous Hall conductivity changes the ellipticity of the electromagnetic waves. When the ellipticity strongly deviates from unity, the linearly polarized waves transform into circularly polarized ones. As shown in Fig. 4(a) and (b), the reflected waves become largely circular for a large window of frequency especially above the band gap. Interestingly, the behavior seems insensitive to the bandwidth. On the other hand, the transmitted waves remain largely linear except in a narrow window of resonance near

band gap and band edge for both flat and dispersive bands [see Fig. 4(c) and (d)]. These results show that topological materials are useful to convert the polarization in opto-electronic devices.

## V. CONCLUSION

Using microscopic pseudo-spin dynamics, we have calculated the full optical conductivity tensor to contrast optical properties of topological flat bands with those of dispersive bands. The anomalous Hall conductivity of flat bands resembles that in the magnetotransport Hall conductivity, in which the sign change appears when the excitation energy matches the cyclotron frequency. It turns out that this sign change is intimately linked with the position of the VHS in the joint denstiy of states. The presence of an anomalous Hall response allows Kerr and Faraday rotations as well as a change of ellipticity. Giant Kerr angles appear below the band gap while the Faraday angle shows a universal value of $\tan^{-1}(\alpha)$ in the dc limit and reaches a maximum absolute value at the band gap. The reflected waves become circularly polarized for excitation energies above the band gap. These results provide a simple characterization technique for topological flat bands and could be useful in opto-electronic devices to generate elliptical polarized light from the linear one. One can introduce correlation effects to the topological flat bands to study fractional quantum Hall (FQH) effects without magnetic field [15, 17, 28, 29]. Flat bands with Chern number greater than one [30] may host a FQH state with even-denominator filling fraction. The Chern number dependence of the optical properties of QAH materials will be an interesting field to study in the future as materials have recently become available [31].

## ACKNOWLEDGMENTS

The authors acknowledge helpful discussions with Christophe de Beule. A.H., J.E., T.L.S., and E.H.H. acknowledge support from the National Research Fund Luxembourg under grants AT-TRACT 7556175, CORE 13579612, and CORE 11352881.

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

## Appendix A: Analytical derivation of the density of states

This Appendix provides additional details on the analytical calculation of the density of states (DOS). We can rewrite Eq. (4) as

$$
\rho(\mathcal{E}) = \sum_{\mathbf{p}} \left[ \delta(\mathcal{E} - |\mathbf{d}(\mathbf{p})|) + \delta(\mathcal{E} + |\mathbf{d}(\mathbf{p})|) \right],
$$
$$
= \frac{a^2}{(2\pi\hbar)^2} \frac{\hbar^2}{a^2} \int dk_x \, dk_y \left[ \delta(\mathcal{E} - |\mathbf{d}|) + \delta(\mathcal{E} + |\mathbf{d}|) \right], \tag{A1}
$$

where $k_x = p_x a/\hbar$ and $k_y = p_y a/\hbar$. By changing the sign of $\mathcal{E}$, i.e. $\mathcal{E} \to -\mathcal{E}$, the second term of the integrand is same as the first term. Therefore, it is sufficient to examine the expression at positive energies. Consequently, the DOS simplifies to

$$
\rho(|\varepsilon|) = \frac{a}{\hbar v} \frac{1}{(2\pi)^2} \int_{-\pi}^{\pi} \int_{-\pi}^{\pi} dk_y \, dk_x \, \delta(|\varepsilon| - D), \tag{A2}
$$

where $\varepsilon = \frac{a\mathcal{E}}{\hbar v}$ and $D = \frac{\sqrt{2}}{4} \sqrt{\left( \cos k_x + \cos k_y \right)^2 + 4}$ are dimensionless energies. To calculate the integral, we employ an orthogonal transformation $(k_x, k_y) \to (\lambda, \xi)$ so that constant coordinate surfaces in the reciprocal space correspond to constant energy surfaces (see Ref. [32] for more details). Explicitly, we define

$$
\lambda = \cos k_x + \cos k_y, \tag{A3}
$$
$$
\tan \xi = \tan \frac{k_y}{2} \cot \frac{k_x}{2}. \tag{A4}
$$

The Jacobian of this transformation is

$$
dk_x \, dk_y = J \, d\xi \, d\lambda, \tag{A5}
$$
$$
J = \frac{1}{\sqrt{1 - \beta \cos^2(2\xi)}}, \qquad \beta = 1 - \frac{\lambda^2}{4}. \tag{A6}
$$

Therefore, the DOS transforms to

$$
\rho(|\varepsilon|) = \frac{1}{4\pi^2} \frac{a}{\hbar v} \int_{-2}^{2} d\lambda \int_{0}^{2\pi} d\xi \, \frac{\delta(|\varepsilon| - D)}{\sqrt{1 - \beta \cos^2(2\xi)}}. \tag{A7}
$$

In order to rewrite the delta function in new coordinates, we employ the inverse transformation

$$
\cos k_x = \frac{\lambda}{2} + \frac{J^{-1} - 1}{\cos(2\xi)}, \tag{A8}
$$
$$
\cos k_y = \frac{\lambda}{2} - \frac{J^{-1} - 1}{\cos(2\xi)}, \tag{A9}
$$

on the delta function as

$$
\delta(\varepsilon - D) = \delta \left( \varepsilon - \frac{\sqrt{2}}{4} \sqrt{\lambda^2 + 4} \right),
$$
$$
= \frac{4|\varepsilon|}{\sqrt{2\varepsilon^2 - 1}} \left[ \delta \left( \lambda - 2\sqrt{2\varepsilon^2 - 1} \right) + \delta \left( \lambda + 2\sqrt{2\varepsilon^2 - 1} \right) \right].
$$

Therefore, the DOS is reduced to

$$
\rho(|\varepsilon|) = \frac{a}{\hbar v} \frac{1}{4\pi^2} \frac{4|\varepsilon|}{\sqrt{2\varepsilon^2 - 1}} \int_{-2}^{2} d\lambda \int_{0}^{2\pi} d\xi \, \frac{\delta \left( \lambda - 2\sqrt{2\varepsilon^2 - 1} \right) + \delta \left( \lambda + 2\sqrt{2\varepsilon^2 - 1} \right)}{\sqrt{1 - \beta\cos^2 2\xi}},
$$
$$
= \frac{a}{\hbar v} \frac{1}{4\pi^2} \frac{4|\varepsilon|}{\sqrt{2\varepsilon^2 - 1}} \, 2 \int_{0}^{2} d\lambda \, \delta \left( \lambda - 2\sqrt{2\varepsilon^2 - 1} \right) 4\mathcal{K}(2 - 2\varepsilon^2),
$$
$$
= \frac{a}{\hbar v} \frac{1}{\pi^2} \frac{8|\varepsilon|}{\sqrt{2\varepsilon^2 - 1}} K(2 - 2\varepsilon^2), \quad \frac{\sqrt{2}}{2} < |\varepsilon| < 1, \tag{A10}
$$

where $\mathcal{K}$ is the elliptic integral of first kind [23], defined as

$$\mathcal{K}(\beta) = \frac{1}{4} \int_0^{2\pi} d\xi \frac{1}{\sqrt{1 - \beta \cos^2 2\xi}}, \quad \beta \leq 1. \tag{A11}$$

The density of states is shown in Fig. 1. Taking into account the two different bands, the integration of the DOS leads to

$$\int_{-\infty}^{\infty} d\mathcal{E} \, \rho(\mathcal{E}) = 2.$$

## Appendix B: Longitudinal conductivity

In this Appendix, we provide additional details about the analytical calculation of the real part of the longitudinal conductivity $\sigma_{xx}$ in Eq. (14). Changing $\omega$ to $\omega + i\eta$ where $\eta \to 0^+$, decomposing as

$$\frac{1}{(\hbar\omega/2)^2 - |\mathbf{d}|^2} = \frac{1}{|\mathbf{d}|} \left( \frac{1}{\hbar\omega - 2|\mathbf{d}|} - \frac{1}{\hbar\omega + 2|\mathbf{d}|} \right),$$

and using the relation $\lim_{\eta \to 0^+} (x + i\eta)^{-1} = \mathcal{P}(1/x) - i\pi\delta(x)$ where $\mathcal{P}$ denotes the principal value, the real part of the longitudinal conductivity can be written as

$$\mathrm{Re}\,\sigma_{xx}(\omega) = \frac{ie^2 v^2}{(2\pi\hbar)^2} \int d^2p \left[ \delta\widetilde{d}_x \frac{\delta\widetilde{d}_x \, (d_z^2 + d_y^2) - \delta\widetilde{d}_y \, (d_x d_y) - \delta\widetilde{d}_z \, (d_x d_z)}{\omega|\mathbf{d}|} + (x \to y, y \to x, z \to z) + (x \to z, y \to y, z \to x) \right]$$

$$\times \left[ \frac{-i\pi \Big( \delta(\hbar\omega - 2|\mathbf{d}|) - \delta(\hbar\omega + 2|\mathbf{d}|) \Big)}{|\mathbf{d}|} \right]$$

$$+ \frac{ie^2 v^2}{(2\pi\hbar)^2} \int d^2p \left[ \delta\widetilde{d}_x \frac{-i\,(\hbar\omega/2)\,d_z\,\delta\widetilde{d}_y + i\,(\hbar\omega/2)\,d_y\,\delta\widetilde{d}_z}{\omega|\mathbf{d}| \left( (\hbar\omega/2)^2 - |\mathbf{d}|^2 \right)} + (x \to y, y \to z, z \to x) + (x \to z, y \to x, z \to y) \right], \tag{B1}$$

where $x \to y$ means that we replace all $x$ indices by $y$: $d_x \to d_y$ and $\delta\widetilde{d}_x \to \delta\widetilde{d}_y$.

Inserting the components of vectors $\mathbf{d}$ and $\delta\mathbf{d}$, the integrand of the second integral in $\mathrm{Re}\,\sigma_{xx}$ is odd in $p_x$ or $p_y$, so the second integral is zero. On the other hand, we need to calculate the first integral only for positive $\omega$ because symmetry relations then make it possible to obtain the results for negative $\omega$. By this assumption, the second delta function in the first integral vanishes for positive $\omega$ and $d$.

Introducing the definition

$$F(k_x, k_y) = \frac{\cos^2 k_x \, (\cos k_x + \cos k_y)^2 - \cos k_x (\cos k_x + \cos k_y)^3 - 4 \, (\cos k_x + \cos k_y) + (\cos k_x + \cos k_y)^2 + 8}{8 \, \widetilde{\omega} \left( (\cos k_x + \cos k_y)^2 + 4 \right)},$$

the real part of $\sigma_{xx}$ reads as

$$\mathrm{Re}\,\sigma_{xx}(\omega) = \frac{e^2}{2h} \int_{-\pi}^{\pi} dk_x \int_{-\pi}^{\pi} dk_y \, F(k_x, k_y) \, \delta(\hbar\widetilde{\omega} - 2|\mathbf{D}|), \tag{B2}$$

where we have used the dimensionless parameters $\widetilde{\omega} = a/v \, \omega$, $\mathbf{D} = a/(\hbar v) \, \mathbf{d}$, $k_x = p_x a/\hbar$ and $k_y = p_y a/\hbar$. To determine $\mathrm{Re}\,\sigma_{xx}$, we employ the transformation $(k_x, k_y) \to (\lambda, \xi)$ introduced in Appendix A. Switching to the new coordinate system, the delta function is given by

$$\delta(\widetilde{\omega} - 2|\mathbf{D}|) = \delta\left( \widetilde{\omega} - \frac{\sqrt{2}}{2}\sqrt{\lambda^2 + 4} \right),$$

$$= \frac{\sqrt{2}\,|\widetilde{\omega}|}{\sqrt{\widetilde{\omega}^2 - 2}} \left[ \delta\left( \lambda - \sqrt{2}\sqrt{\widetilde{\omega}^2 - 2} \right) + \delta\left( \lambda + \sqrt{2}\sqrt{\widetilde{\omega}^2 - 2} \right) \right]. \tag{B3}$$

Therefore, we rewrite $\mathrm{Re}\,\sigma_{xx}$ in new coordinates $(\lambda, \xi)$ as

$$\mathrm{Re}\,[\sigma_{xx}(\omega)] = \frac{e^2}{16h} \int_{-2}^{2} d\lambda \int_0^{2\pi} d\xi \, \frac{1}{\sqrt{1 - \beta \cos^2 2\xi}} \, F(\lambda, \xi) \frac{\sqrt{2}\,|\widetilde{\omega}|}{\sqrt{\widetilde{\omega}^2 - 2}} \Big[ \delta\left( \lambda - \Omega \right) + \delta\left( \lambda + \Omega \right) \Big], \tag{B4}$$

$$F(\lambda, \xi) = \frac{\left( \frac{\lambda}{2} + \frac{\sqrt{1 - \beta\cos^2 2\xi} - 1}{\cos 2\xi} \right)^2 \lambda^2 - \left( \frac{\lambda}{2} + \frac{\sqrt{1 - \beta\cos^2 2\xi} - 1}{\cos 2\xi} \right) \lambda^3 - 4\lambda + \lambda^2 + 8}{8 \, \widetilde{\omega} \, (\lambda^2 + 4)}, \tag{B5}$$

where $\beta = 1 - \lambda^2/4 = 2 - \widetilde{\omega}^2/2$ and $\Omega = \sqrt{2\left(\frac{a\omega}{v}\right)^2 - 4}$. Performing the integrations over $\lambda$ and $\xi$, we find

$$\mathrm{Re}\,\sigma_{xx}(\omega) = \frac{4\mathcal{K}\left(2 - \frac{\widetilde{\omega}^2}{2}\right) - 2\left(\widetilde{\omega}^2 - 2\right)\mathcal{L}\left(2 - \frac{\widetilde{\omega}^2}{2}\right)}{\sqrt{2\widetilde{\omega}^2}\sqrt{\widetilde{\omega}^2 - 2}}, \qquad \sqrt{2} < \widetilde{\omega} < 2, \tag{B6}$$

where $\mathcal{K}$ is defined in Eq. (A11) and $\mathcal{L}$ is the elliptic integral of the second kind

$$\mathcal{L}(\beta) = \frac{1}{4}\int_0^{2\pi} d\xi\sqrt{1 - \beta\cos^2 2\xi}, \qquad \beta \leq 1. \tag{B7}$$

Note that the real component of the longitudinal conductivity is zero if $\omega < \frac{\sqrt{2}v}{a}$ and $\omega > \frac{2v}{a}$.

## Appendix C: Hall conductivity

In this appendix, we are going to present the detailed derivation of the analytic form of the Hall conductivity. To this end, in the following, we calculate the real and imaginary part, individually.

### 1. Imaginary part

Let us first calculate the imaginary part of the Hall conductivity which is given by

$$\begin{aligned}
\mathrm{Im}\,\sigma_{yx}(\omega) = {}& \frac{ie^2v^2}{(2\pi\hbar)^2}\int d^2p\left[\left(\delta\widetilde{d}_y\,\frac{\delta\widetilde{d}_y\,(-i\hbar\omega/2)\,d_z + \delta\widetilde{d}_z(i\hbar\omega/2)\,d_y}{\omega|\mathbf{d}|} + (x\to y, y\to z, z\to -z)\right)\right.\\
&\left.+ b\sin(\frac{p_y a}{\hbar})\,\frac{\delta\widetilde{d}_x\,(-i\hbar\omega/2)\,d_y + \delta\widetilde{d}_y(i\hbar\omega/2)\,d_x}{\omega|\mathbf{d}|}\right]\\
&\times\left[\frac{-i\pi\Big(\delta(\hbar\omega - 2|\mathbf{d}|) - \delta(\hbar\omega + 2|\mathbf{d}|)\Big)}{|\mathbf{d}|}\right]\\
&+ \frac{ie^2v^2}{(2\pi\hbar)^2}\int d^2p\left[\left(\delta\widetilde{d}_y\,\frac{\delta\widetilde{d}_x\,(d_z^2 + d_y^2) - \delta\widetilde{d}_y\,(d_x d_y) - \delta\widetilde{d}_z\,(d_x d_z)}{\omega|\mathbf{d}|\left((\hbar\omega/2)^2 - |\mathbf{d}|^2\right)} + (x\to y, y\to x, z\to z)\right)\right.\\
&\left.+ b\sin\left(\frac{p_y a}{\hbar}\right)\,\frac{\delta\widetilde{d}_z\,(d_x^2 + d_y^2) - \delta\widetilde{d}_x\,(d_x d_z) - \delta\widetilde{d}_y\,(d_y d_z)}{\omega|\mathbf{d}|\left((\hbar\omega/2)^2 - |\mathbf{d}|^2\right)}\right],
\end{aligned} \tag{C1}$$

where, as before, $z \to -z$ denotes $d_z \to -d_z$ and $\delta\widetilde{d}_z \to -\delta\widetilde{d}_z$. Since the integrand in the second term of $\mathrm{Im}\,\sigma_{yx}$ is odd with respect to $p_x$ or $p_y$, the second integral is zero. By a change of variables $(k_x, k_y)$ to $(\lambda, \xi)$ as mentioned before in Appendix A, we find

$$\begin{aligned}
\mathrm{Im}\,\sigma_{yx} = {}& -\frac{e^2}{h}\frac{1}{8}\int_{-2}^2 d\lambda\int_0^{2\pi} d\xi\,\frac{1}{\sqrt{1 - \beta\cos^2 2\xi}}\frac{(\lambda - 2)(\lambda + 2)}{(\lambda^2 + 4)}\\
&\times\frac{|\widetilde{\omega}|}{\sqrt{\widetilde{\omega}^2 - 2}}\left[\delta\left(\lambda - \sqrt{2}\sqrt{\widetilde{\omega}^2 - 2}\right) + \delta\left(\lambda + \sqrt{2}\sqrt{\widetilde{\omega}^2 - 2}\right)\right],
\end{aligned} \tag{C2}$$

where the delta functions in the new coordinate system is shown in Eq. (B3). Performing the integrations over $\lambda$ and $\xi$ gives

$$\mathrm{Im}\,\sigma_{yx}(\omega) = \frac{e^2}{h}\frac{4 - \widetilde{\omega}^2}{\widetilde{\omega}}\frac{1}{\sqrt{\widetilde{\omega}^2 - 2}}\,\mathcal{K}\left(2 - \frac{\widetilde{\omega}^2}{2}\right), \qquad \sqrt{2} < \widetilde{\omega} < 2. \tag{C3}$$

## 2. Real part

The real part of $\sigma_{yx}(\omega)$ is given by

$$
\begin{aligned}
\mathrm{Re}\,\sigma_{yx}(\omega) = {}&\frac{ie^2v^2}{(2\pi\hbar)^2}\int d^2p\left[\left(\delta\widetilde{d}_y\,\frac{\delta\widetilde{d}_x\,(d_z^2+d_y^2)-\delta\widetilde{d}_y\,(d_xd_y)-\delta\widetilde{d}_z\,(d_xd_z)}{\omega|\mathbf{d}|}+(x\to y,y\to x,z\to z)\right)\right.\\
&\left.+\,b\sin(\frac{p_ya}{\hbar})\,\frac{\delta\widetilde{d}_z\,(d_x^2+d_y^2)-\delta\widetilde{d}_x\,(d_xd_z)-\delta\widetilde{d}_y\,(d_yd_z)}{\omega|\mathbf{d}|}\right]\\
&\times\left[\frac{-i\pi\Big(\delta(\hbar\omega-2|\mathbf{d}|)-\delta(\hbar\omega+2|\mathbf{d}|)\Big)}{|\mathbf{d}|}\right]\\
+{}&\frac{ie^2v^2}{(2\pi\hbar)^2}\int d^2p\left[\left(\delta\widetilde{d}_y\,\frac{-i\,(\hbar\omega/2)\,d_z\,\delta\widetilde{d}_y+i\,(\hbar\omega/2)\,d_y\,\delta\widetilde{d}_z}{\omega|\mathbf{d}|\left((\hbar\omega/2)^2-|\mathbf{d}|^2\right)}+(x\to y,y\to x,z\to-z)\right)\right.\\
&\left.+\,b\sin(\frac{p_ya}{\hbar})\,\frac{-i\,(\hbar\omega/2)\,d_y\,\delta\widetilde{d}_x+i\,(\hbar\omega/2)\,d_x\,\delta\widetilde{d}_y}{\omega|\mathbf{d}|\left((\hbar\omega/2)^2-|\mathbf{d}|^2\right)}\right]. \qquad\text{(C4)}
\end{aligned}
$$

The first term of $\mathrm{Re}\,\sigma_{yx}$, including delta functions, is odd, and consequently the first integral is zero. By substitution of the components of $\mathbf{d}$ and $\delta\mathbf{d}$, the real part can be expressed as

$$
\mathrm{Re}\,\sigma_{yx}(\omega) = \frac{e^2}{h}\frac{1}{4\pi}\int dk_x\,dk_y\,\frac{(\cos k_x+\cos k_y-2)\,(\cos k_x+\cos k_y+2)}{\left(2\big(\frac{a\omega}{v}\big)^2-4-(\cos k_x+\cos k_y)^2\right)\sqrt{(\cos k_x+\cos k_y)^2+4}}. \qquad\text{(C5)}
$$

In order to perform the integration, we map $(k_x, k_y)$ space to $(\lambda, \xi)$ space as explained in Appendix A. Therefore, the real part of the Hall conductivity in new space is rewritten as

$$
\mathrm{Re}\,\sigma_{yx}(\omega) = \frac{e^2}{h}\frac{1}{4\pi}\int_{-2}^{2}d\lambda\int_{0}^{2\pi}d\xi\,\frac{1}{\sqrt{1-\beta\cos^2 2\xi}}\frac{(\lambda-2)\,(\lambda+2)}{(\Omega^2-\lambda^2)\sqrt{\lambda^2+4}}. \qquad\text{(C6)}
$$

Performing the integration with respect to $\xi$, we obtain

$$
\mathrm{Re}\,\sigma_{yx}(\omega) = \frac{e^2}{h}\frac{1}{\pi}\int_{-2}^{2}d\lambda\,\frac{(\lambda-2)\,(\lambda+2)}{\left(2\big(\frac{a\omega}{v}\big)^2-4-\lambda^2\right)\sqrt{\lambda^2+4}}\,\mathcal{K}\left(1-\frac{\lambda^2}{4}\right), \qquad\text{(C7)}
$$

We performed the remaining integration over $\lambda$ numerically. Note that for $\omega=0$, we recover the result that $\mathrm{Re}\,\sigma_{yx}(\omega)=e^2/h$.