# Peer review of "Kerr and Faraday rotations in topological flat and dispersive band structures"

_SciPost Physics_

## Round 2 · Referee Report · Anonymous · 2021-9-23

Report

The authors have improved the title, abstract, introduction, and conclusion. In turn, their motivation and main arguments are easier to follow, contextualized, and clearly separated from the existing literature. The spelling, grammar and formatting has also been tidied up, and connections to future work have been established. I now recommend this paper for publication in SciPost Physics.

There are a couple of optional points that the authors may wish to consider:

-- You have discussed how one can tune from QAH to IQH physics by flattening the dispersive topological bands. Would it be possible to do the converse? i.e. modify an IQH system to uniquely recover QAH resonant profiles? In practice, do you expect a finite transition region with respect to band gap and VHS energy?

-- You mention how this technique can be used to characterize topological flat bands, which is currently an experimental challenge. To what extent can the bandwidth be "characterized" using Kerr and Faraday rotations? i.e. to what resolution and under which conditions? Proximity of bands to Fermi energy is not an issue (as with ARPES)?

Typo: denstiy --> density

  • validity: good
  • significance: good
  • originality: good
  • clarity: good
  • formatting: good
  • grammar: good

Author:  Eddwi Hasdeo  on 2021-10-20  [id 1869]

(in reply to Report 1 on 2021-09-23)
Category:
remark
answer to question

Dear Referee,

Thank you very much for your fair assessment to our paper and recommendation for publication.
In the following we answer your question.
Q1: You have discussed how one can tune from QAH to IQH physics by flattening the dispersive topological bands. Would it be possible to do the converse? i.e. modify an IQH system to uniquely recover QAH resonant profiles? In practice, do you expect a finite transition region with respect to band gap and VHS energy?

A1: Thank you very much for your interesting question.
No, if we limit ourselves in 2D systems. QAH system has energy dispersions where curvatures of valence and conduction bands are opposite to each other. Such dispersions are not possible to be realized in IQH systems whose Hamiltonian mimics the harmonic oscillator. In 3D systems, however, it is possible to have dispersive Landau levels in which momentum k parallel to B field remains a good quantum number. A well-known example is the Landau levels of Weyl semimetals. However, this study is beyond the scope of our project.

Q2: You mention how this technique can be used to characterize topological flat bands, which is currently an experimental challenge. To what extent can the bandwidth be "characterized" using Kerr and Faraday rotations? i.e. to what resolution and under which conditions?

A2: Thank you very much for your question.
The Kerr and Faraday effects are not the best methods to observe the bandwidth because Kerr and Faraday effect are mostly determined by Re(sigma_xy) which shows finite value even for frequency above the band width. The easiest way to measure the bandwidth is by using optical absorption spectra which is proportional to Re(\sigma_xx). The Kerr and Faraday rotations are meant to observe the anomalous Hall effect that originates from the band topology.

Q3: Proximity of bands to Fermi energy is not an issue (as with ARPES)?
A3: In ARPES, the substrate needs to be metallic thus it is difficult to tune the Fermi energy.
In optical measurement, such a tuning of Fermi energy is possible since the substrate can be insulators. Thus proximity of bands to Fermi energy is not an issue in optical measurement.

We will add additional remarks related to Q1-Q3 and fix the typo during the proof of the manuscript.

---

## Round 2 · Referee Report · Anonymous · 2021-11-3

Report

In this paper, the authors studied the ac response function of quantum anomalous Hall states vs. integer quantum Hall states. In IQH, there are resonance and sign changing features at the cyclotron frequency, while in QAH, these two do not necessarily show up at the same energy. By studying a model of QAH with a tunable parameter controlling the flatness of the band, the authors found that this is because generically when a band is not flat, the band gap and the Van Hove Singularity do not necessarily show up at the same energy. While the analysis is clear and I believe the result to be correct, I do not find it very surprising. Generically, one would expect that band gap and Van Hove Singularity to each lead to some nontrivial features in the response function and in a dispersive band, they do not have to be at the same energy. The authors also talked about using optical features to characterize topological flat bands. Using Kerr and Faraday rotation to characterize topological bands is not a new idea. The proposal in the paper might be focused on the flatness of the band but I find this part very vague. It is not clear quantitatively what feature can be characterized by the optical probes. Is it the total bandwidth of the band? Or the distance between band gap and Van Hove Singularities? In fact, as shown in figure 3 and 4, the probes may not even have a very strong response to Van Hove Singularities.

Because of all this, I do not think the paper contains enough new physics to be published in Scipost.

  • validity: high
  • significance: low
  • originality: ok
  • clarity: high
  • formatting: excellent
  • grammar: perfect

Author:  Eddwi Hasdeo  on 2021-11-04  [id 1911]

(in reply to Report 2 on 2021-11-03)
Category:
answer to question

Dear Referee,
Thank you very much for your time to review our manuscript. Our motivation is first to study the optical properties of topological flat bands and next to provide a simple way to characterize this system.
So far, characterization of bandwidth is limited only via ARPES, which might not be accessible to everyone and does not give information about topology of the system. Using optical response, we can characterize the bandwidth and bandgap from optical absorption (Re(sigma_xx)). Additionally, Re (sigma_xy) provides the topological nature of the system and can confirm the flatness of the band by looking at the sign change of Re (sigma_xy) at the band gap.
The Re(sigma_xy) can be probed using the Faraday rotations.

We believe that our work is original and meets the standard of SciPost Physics.
In the proof, We will make clearer the role of Faraday and Kerr rotations as complementary tools to optical absorption for characterizing the topology and band shapes (band width and band gap).

Anonymous on 2021-12-10  [id 2024]

(in reply to Eddwi Hasdeo on 2021-11-04 [id 1911])

I would like to thank the authors for their reply, but I don't think it sufficiently addresses my concern. In particular, the authors' description of how to use optical probes to characterize topological flat band remains vague. The statement consists of a few sentences with very minimal quantitative details on the relation between what is measured (optical response) and what we want to know (characteristics of flat topological band). Moreover, the author did not address my question about the lack of feature at van Hover singularities. Because of this, I do not think this paper is suitable for publication in SciPost.

---

## Round 2 · Author Response

Dear Editor,

We revised the manuscript according to the Referee's suggestions, especially we change our title to be more specific: "Kerr and Faraday rotations of topological flat and dispersive bands"; we rewrite Introduction to express the motivation and novelty of this paper; and we fix all minor details. We believe that the revised version has addressed all Referee criticism and hopefully suitable for publication in Scipost Physics.
We take a chance to reemphasize the motivation of this paper. Magnetic field and Berry curvature are dual to each other. All phenomena occurring in quantum Hall (QH) states with magnetic field, can also be seen in quantum anomalous Hall (QAH) states without magnetic field (with Berry curvature), e.g. non-dissipative chiral edge modes and quantized dc Hall conductance. However the dynamical (ac) Hall conductivity of in QH states is distinct from the dispersive QAH band. We resolve this different profiles by treating topological flat and dispersive bands in an equal footing. In the limit of small band width, the Hall conductivity of topological flat bands resembles that of flat Landau levels in quantum Hall states with the sign change signature near the resonance. The comparison with Hall conductivity for topological dispersive bands can serve as a simple tool to characterize the topological flat bands. Please see below point-by-point answers of your comments.

---

## Round 2 · List of Changes

1. Title
2. Abstract
3. Introduction
4. Outlook for future works at the end of the conclusion.

---

## Editorial Decision

editor-in-charge_assigned